

# Authentication of Jarrah (*Eucalyptus marginata*) honey through its nectar signature and assessment of its typical physicochemical characteristics

Md Khairul Islam[1,2], Elizabeth Barbour[2,3] and Cornelia Locher[1,2]

[1] Division of Pharmacy, School of Allied Health, University of Western Australia, Perth, Western Australia, Australia
[2] CRC for Honey Bee Products, Yanchep, Western Australia, Australia
[3] School of Agriculture and Environment, University of Western Australia, Perth, Western Australia, Australia

## ABSTRACT

Jarrah (*Eucalyptus marginata*) is a dominant forest tree endemic to the southwest of Western Australia. Its honey is appreciated for its highly desirable taste, golden colour, slow crystallisation, and high levels of bioactivity, which have placed Jarrah in the premium product range. However, whilst customers are willing to pay a high price for this natural product, there is currently no standard method for its authentication. As honey is naturally sourced from flower nectar, a novel route of authentication is to identify the nectar signature within the honey. This study reports on a high-performance thin layer chromatography (HPTLC)-based authentication system which allows the tracing of six key marker compounds present in Jarrah flower nectar and Jarrah honey. Four of these markers have been confirmed to be epigallocatechin, lumichrome, taxifolin and o-anisic acid with two (Rf 0.22 and 0.41) still chemically unidentified. To assist with the characterisation of Jarrah honey, a range of physicochemical tests following Codex Alimentarius guidelines were carried out. A blend of authenticated Jarrah honey samples was used to define the properties of this honey type. The blend was found to have a pH of 4.95, an electric conductivity of 1.31 mS/cm and a moisture content of 16.8%. Its water-insoluble content was 0.04%, its free acidity 19 milli-equivalents acid/kg and its diastase content 13.2 (DN). It also contains fructose (42.5%), glucose (20.8%), maltose (1.9%) and sucrose (<0.5%). The HPTLC-based authentication system proposed in this study has been demonstrated to be a useful tool for identifying Jarrah honey and might also act as a template for the authentication of other honey types.

# INTRODUCTION

Honey is a naturally sweet substance produced by honeybees (*Apis mellifera*) from gathered flower nectar or insect exudate (*Alimentarius, 2017*; *Garcia-Seval et al., 2022*). Jarrah honey is sourced from the flower nectar of *Eucalyptus marginata* trees, which are dominant

Corresponding author
Md Khairul Islam,
khairul.islam@uwa.edu.au

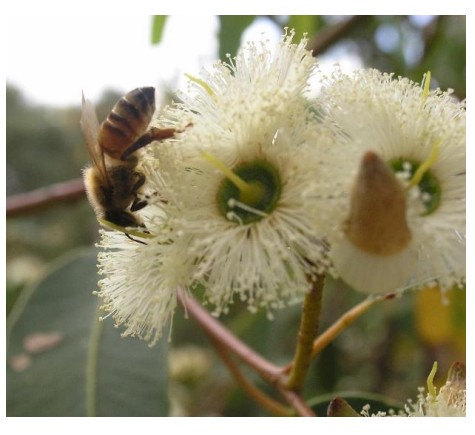
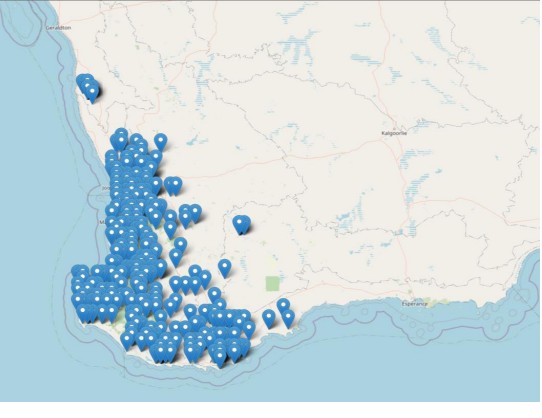

a

b

**Figure 1** **(A) Jarrah (*Eucalyptus marginata*) flowers being visited by a honeybee. Photo source credit: Linda Manning, BICWA. (B) The distribution of the species through the southwest of Western Australia.** Image source credit: Image used with the permission of the Western Australian Herbarium, Department of Biodiversity, Conservation and Attractions (https://florabase.dbca.wa.gov.au/help/copyright).

throughout the southwest of Western Australia and grow slowly over more than 100 years to eventually reach a height of 40 m (*Dixon et al., 2022*; *Hingston, Dimmock & Turton, 1980*). They are stringybark eucalypts with distinctive striations in their silvery bark and produce white to creamy flowers (Fig. 1A) every 3–4 years depending on the weather cycle (*Goebel, 1944*; *Robinson & Williams, 2011*). Spring flowering, which occurs between September and January, starts in the southern forests and moves to the north across the species range as the flowering season progresses (*Coleman, 1962*; *Specht, Hopkins & AJM, 1981*) (Fig. 1B).

Jarrah honey is one of many monofloral honey types harvested from Western Australia (*Cokcetin, 2015*). It is a rare and unique commodity because Jarrah forests are endemic to the southwest of Western Australia (Fig. 1) (*Department of Biodiversity, 2002*). The honey is popular due to its distinct flavour profile, deep golden colour, and high levels of bioactivity (*Chandler et al., 1974*; *Lawag et al., 2023*). As a so-called peroxide honey, its antibacterial properties are mainly associated with the enzymatic activity of glucose oxidase, producing hydrogen peroxide (*Guttentag et al., 2021a*; *Guttentag et al., 2021b*; *Irish, Blair & Carter, 2011*; *Manning, 2011*). It also features high levels of phenolic constituents which are also linked to its antioxidant activity (*Lawag et al., 2023*). Another typical feature of Jarrah honey is its low tendency to crystallise which is caused by its high fructose level relative to glucose and its generally low pollen count which does not provide 'seeds' to initiate crystallisation (*Islam et al., 2022*; *Ji et al., 2023*). Due to its high fructose content, Jarrah honey also has a lower glycaemic index (54 ± 3 GI value) compared to many other kinds of honey (*Bogdanov, 2012*; *Dawes & Dall, 2014*) and contains non-digestible carbohydrates that promote prebiotic activity (*Cokcetin, 2015*).

As Jarrah honey is a premium product, similar to New Zealand Manuka honey, authentication is vital to protect consumers from mislabelling or adulteration (*Lin et al.,*

*2017*; *McDonald et al., 2018*). Melissopalynology, which is a commonly employed honey authentication method, has proven insufficient for Jarrah honey due to its production in native bushland and forests where co-flowering of other plant species is a common occurrence. The quality of Jarrah pollen to support bee health is good (*Coleman, 1962*), but the pollen lacks two essential amino acids (isoleucine and histidine) (*Manning, 2001*), which bees obtain from the pollen of other species flowering alongside Jarrah trees. This bee behaviour can undermine melissopalynology as an authentication method. Therefore pollen found within Jarrah honey is argued to denote the biogeographical region of its production (*Louveaux, Maurizio & Vorwohl, 1970*; *Newstrom-Lloyd, 2017*; *Newstrom-Lloyd, Raine & Li, 2017*; *Sniderman et al., 2018*) rather than its monofloral status (*Islam et al., 2022*).

To date the pursuit of the authentication of Jarrah honey has been an elusive goal. This article offers a novel approach through the investigation of the non-sugar components found in Jarrah honey as well as Jarrah flower nectar. Using high-performance thin layer chromatography (HPTLC), a typical honey signature can be determined that is directly traceable to the honey's floral nectar source. This authentication approach allows, for the first time, to scientifically characterise typical physicochemical characteristics of Jarrah honey, as well as assist governing bodies to establish new standards by which uniformed honey types can be identified for accurate labelling.

## MATERIALS AND METHODS
### Chemicals and reagents

Portions of this text were previously published as part of a thesis (*Islam, 2022*). Chemicals and reagents used in this study and their suppliers: fructose, maltose, aniline, vanillin and sodium bisulfite (Sigma-Aldrich, St. Louis, MO, USA), 4,5,7-trihydroxyflavanone (Alfa Aesar, England, UK), anhydrous sodium sulfate and boric acid (Pharma Scope, Welshpool, WA, Australia), glucose, sucrose, sodium carbonate anhydrous, sodium hydroxide and zinc acetate dihydrate (Chem- Supply Pty Ltd., St. Gillman, SA, Australia), gallic acid, diphenylamine, phosphoric acid and phenolphthalein (Ajax Finechem Pvt Ltd., Sydney, Australia), potassium ferricyanide (Rowe Scientific, Wangara, WA, Australia).

Solvents used in this study and their suppliers: methanol (Scharlau, Barcelona, Spain), dichloromethane (Merck KGaA, Darmstadt, Germany), ethanol, ethyl acetate and formic acid (Ajax Finechem Pvt Ltd., Sydney, Australia), 1-butanol and glacial acetic acid (Chem-Supply Pty Ltd., St. Gillman, SA, Australia), 2-propanol and toluene (Asia Pacific Specialty Chemicals Ltd, Sydney, Australia).

The phenolic standards were sourced from: kojic acid, o-anisic acid (2-methoxybenzoic acid), 2,3,4-trihydroxy benzoic acid and epigallocatechin gallate (Angene International Ltd. Nanjing, China), lumichrome (Sigma Aldrich, Castle Hill, NSW, Australia), hesperetin and m-coumaric acid (Combi-Blocks Inc., San Diego, CA, USA) and taxifolin (AK Scientific, Inc., Union City, CA, USA).

Silica gel 60 $F_{254}$ HPTLC glass plates (20 cm × 10 cm) were purchased from Merck KGaA, Darmstadt, Germany.

## Sample collection and preparation
### Jarrah nectar

Jarrah nectar was collected from Jarrah (*Eucalyptus marginata*) trees within their known range (Fig. 1B), specifically in Jarrahdale (32.33814°S, 116.06267°E; Date: 29 October 2023), which is located on the Darling escarpment within the Jarrah Forest biogeographical region, and from Margaret River (33.96226°S, 115.02576°E; Date: 14 October 2023) within the Warren region (*Department of Agriculture, Water and the Environment, Commonwealth of Australia, 2012*). The trees for nectar collection were identified by their flowers, buds and capsules together with their distinct bark (*Dell, Havel & Malajczuk, 2012*; *Hingston, O'Connell & Grove, 1989*; *Slee et al., 2020*). Flowers were obtained in the early morning, stored in a sealed container, and placed in a fridge for transport to the laboratory. Each flower was washed three times with 10 µl of distilled water and the solution collected in an opaque vial, which was stored at 4 °C until analysis.

### Jarrah honey

From the southwest of Western Australia 500 samples were collected and a subsample of 31 samples labelled as Jarrah honey (Table 1) were provided by beekeepers between November 2015 and December 2020 from apiary sites within the Jarrah Forest. The samples were stored in glass jars in the dark at room temperature until analysis.

### Jarrah honey blend

Thirteen samples were selected as "typical" Jarrah honey samples. A sub-sample of 5 g was batched and mixed to create a blend ('HPTLC analysis of organic honey and nectar extracts').

## High-performance thin layer chromatography (HPTLC)

All samples were analysed by HPTLC for their non-sugar components. This technique was selected as it offered a quick detection of the respective nectar and honey signatures through derivatisation and visualisation at different wavelengths, as well as the ability to identify and quantify constituents of interest.

## Sample preparation
### Preparation of honey and nectar samples

For the preparation of nectar and honey organic extracts of approximately 1 g of honey, or a nectar solution, were mixed with two mL of deionised water. The aqueous solution was then extracted three times with five mL of dichloromethane. The combined organic extracts were dried with anhydrous $MgSO_4$, filtered and the solvent evaporated at ambient temperature. The extract was stored at 4 °C and reconstituted in 100 µL dichloromethane before HPTLC analysis.

For the sugar analysis, 100 mg of honey was dissolved in 80 mL of 50% aqueous methanol and then made up to 100 mL with 50% aqueous methanol.

### Standard, mobile phase and reagent preparation

For the HPTLC analysis of the organic honey and nectar extracts, a methanolic solution of 0.5 mg/mL of 4,5,7-trihydroxyflavanone was prepared as a reference standard and a

**Table 1  Coded honey samples showing the beekeeper-identified honey source, sub-biogeographical region and harvest date.**

| Code | Honey type (as identified by the beekeeper) | Biogeographical Sub-region | Collection Period |
|---|---|---|---|
| JAR-009 | Jarrah | Northern Jarrah Forest | Nov 2015 |
| JAR-050 | Jarrah | Northern Jarrah Forest | Feb 2018 |
| JAR-066 | Jarrah, Wattle | Southern Jarrah Forest | Nov 2017 |
| JAR-068 | Jarrah | Northern Jarrah Forest | Jan 2017 |
| JAR-069 | Jarrah | Northern Jarrah Forest | Jan 2017 |
| JAR-075 | Jarrah | Northern Jarrah Forest | Dec 2016 |
| JAR-078 | Jarrah | No Data | No Data |
| JAR-165 | Jarrah | No Data | No Data |
| JAR-169 | Jarrah | Northern Jarrah Forest | Dec 2017 |
| JAR-172 | Jarrah | Northern Jarrah Forest | Jan 2018 |
| JAR-234 | Jarrah | Swan Coastal Plain Perth/ Northern Jarrah Forest | No Data |
| JAR-258 | Jarrah | Southern Jarrah Forest | Nov 2019 |
| JAR-263 | Jarrah | No Data | No Data |
| JAR-265 | Jarrah | Northern Jarrah Forest | Dec 2019 |
| JAR-266 | Jarrah | Northern Jarrah Forest | Dec 2019 |
| JAR-270 | Jarrah | No Data | No Data |
| JAR-287 | Jarrah, Wildflower | Northern Jarrah Forest/ Southern Jarrah Forest | Dec 2019 |
| JAR-295 | Jarrah, Blackbutt (Swan River) | Northern Jarrah Forest | Feb 2020 |
| JAR-298 | Jarrah | Swan Coastal Plain Perth/ Northern Jarrah Forest | Nov 2019 |
| JAR-302 | Jarrah | Swan Coastal Plain Perth | Jan 2019 |
| JAR-313 | Jarrah | Northern Jarrah Forest | Jan 2015 |
| JAR-328 | Jarrah | Southern Jarrah Forest | Nov 2019 |
| JAR-338 | Jarrah | Northern Jarrah Forest | Jan 2020 |
| JAR-340 | Jarrah | Northern Jarrah Forest | Jan 2020 |
| JAR-342 | Jarrah | Northern Jarrah Forest | Jan 2020 |
| JAR-344 | Jarrah | Northern Jarrah Forest | Jan 2020 |
| JAR-348 | Jarrah | Northern Jarrah Forest | Feb 2020 |
| JAR-364 | Jarrah | Northern Jarrah Forest/ Southern Jarrah Forest | No Data |
| JAR-367 | Jarrah, Avocado | Northern Jarrah Forest/ Southern Jarrah Forest | Dec 2019 |
| JAR-373 | Jarrah | Northern Jarrah Forest/ Southern Jarrah Forest | No Data |
| JAR-402 | Jarrah | Northern Jarrah Forest | Dec 2020 |

mixture of toluene: ethyl acetate: and formic acid (6:5:1, v/v/v) as mobile phase. The vanillin derivatisation reagent was prepared by dissolving 1 g of vanillin in 100 mL of ethanol, followed by the dropwise addition of two mL of sulphuric acid.

To identify and quantify the honey's main sugars, standard glucose, fructose, maltose, and sucrose solutions were prepared by dissolving the respective sugar in 50% aqueous methanol. The concentration of standards was fructose (250 ng/$\mu$L), glucose (250 ng/$\mu$L), maltose (50 ng/$\mu$L) and sucrose (100 ng/$\mu$L).

A 3:5:1 $v/v/v$ mixture of 1-butanol: 2-propanol: boric acid (5 mg/mL in water) was prepared as a mobile phase. For the derivatisation reagent, 2 g of diphenylamine and two mL of aniline were dissolved in 80 mL of methanol. After the addition of 10 mL of phosphoric acid (85%), the solution was made up to 100 mL using methanol.

For the determination of 5-hydroxymethylfurfural (HMF) content, Carrez solution I was prepared by dissolving 15 g potassium ferrocyanide ($K_4Fe(CN)_6 \cdot 3H_2O$) in 100 ml deionized water. Carrez solution II was prepared by dissolving 30 g zinc acetate ($Zn(CH_3CO_2)_2 \cdot 2H_2O$) in 100 mL deionized water.

## Sample analysis
### Jarrah nectar and jarrah honey organic extracts
The reference standard (4 $\mu$L) and the respective nectar (5 uL and 20 uL) or honey (5 uL) organic extract solutions were applied as eight mm bands at eight mm from the lower edge of the HPTLC plate (glass plates 20 × 10 cm, silica gel 60 F254) at a rate of 150 nLs$^{-1}$ using a semi-automated HPTLC application device (Linomat 5; CAMAG). The chromatographic separation was performed in a saturated and activated (33% relative humidity) automated development chamber (ADC2; CAMAG). The plates were pre-conditioned with the mobile phase for 5 min and automatically developed to a distance of 70 mm at a fixed ambient temperature (*Islam, 2022*).

The obtained chromatographic results were recorded using an HPTLC imaging device (TLC Visualizer 2; CAMAG) at 254 nm and 366 nm, respectively. After the initial documentation of the chromatographic results, each plate was derivatised with three mL of vanillin-sulfuric acid reagent and heated for 3 min at 115 °C using a CAMAG TLC Plate Heater III. The plate was cooled to room temperature and analysed with the HPTLC imaging device under white light and at 366 nm. The scanning of individual major bands in the nectar and honey extracts, before and after derivatisation, was carried out using a TLC Scanner 4. The chromatographic images were digitally processed and analysed using specialised HPTLC software (visionCATS; CAMAG), which was also used to control the individual instrumentation modules (*Islam et al., 2021a*; *Locher et al., 2018*).

### Jarrah honey sugar profile
Sugar standard solutions were applied as eight mm bands at eight mm from the lower edge of the HPTLC plate (glass plates 20× 10 cm, silica gel 60 F254) at a rate of 50 nLs$^{-1}$ using a semi-automated HPTLC application device (Linomat 5; CAMAG). To prepare the glucose, fructose, sucrose and maltose standard curves, 1 $\mu$L, 2 $\mu$L, 3 $\mu$L, and 4 $\mu$L of the respective standard solutions were applied. To quantify the fructose and glucose content, and maltose and sucrose content of the Jarrah honey blend, 2 $\mu$L and 6 $\mu$L respectively, of the honey solution was applied.

The chromatographic separation was performed in a saturated (33% relative humidity) automated development chamber (ADC2; CAMAG). The development chamber was

saturated for 60 min, and the plates were pre-conditioned with the mobile phase for 5 min, automatically developed to a distance of 85 mm at a fixed ambient temperature and dried for 5 min. The obtained chromatographic results were recorded using an HPTLC imaging device (TLC Visualizer 2; CAMAG) under white light (*Islam, 2022*).

After the initial documentation of the chromatographic results, each plate was derivatised with two mL of aniline-diphenylamine-phosphoric acid reagent using a TLC derivatiser (CAMAG Derivatiser). The derivatised plates were heated for 10 min at 115 °C using a CAMAG TLC Plate Heater III. The plates were then cooled to room temperature and analysed with the HPTLC imaging device under white light (*Islam et al., 2020*; *Islam et al., 2021b*). The chromatographic images were digitally processed and analysed using specialised HPTLC software (visionCATS; CAMAG), which was also used to control the individual instrumentation modules.

### Physico-chemical characteristics of jarrah honey

The key physicochemical characteristics such as pH, electric conductivity, Brix value and moisture content, water-insoluble content, free acidity, diastase and 5-hydroxymethylfurfural (HMF) content were determined by standard analytical methodologies following Codex Alimentarius guidelines (*Alimentarius, 2017*).

In brief, the pH of honey was measured by dissolving 10 g of honey in 75 ml of carbon-dioxide-free water (*Meda et al., 2005*) and the resulting pH of the solution was determined with a calibrated pH meter (HI98131; Hanna Instruments, Woonsocket, RI, USA).

The electrical conductivity of a 20% (w/v) honey solution was measured at 22 °C using an Electrical Conductometer (HI98131, Hanna Instruments, Rhode Island, USA) and expressed as milliSiemens per centimetre (mS/cm) (*Adaškevičiūte et al., 2019*).

Brix value and moisture content were determined simultaneously by spreading the honey over the entire surface of the reading window of a digital refractometer (HI96800, Hanna Instruments, Woonsocket, RI, USA). The moisture content of the honey sample was derived from the respective 'Refractive index (20 °C) *vs* Moisture content (percent)' chart (*Bogdanov, Martin & Lullmann, 2002*).

Water insoluble content was determined by dissolving 10 g of honey into deionised water. The solution was filtered through a previously dried and weighed filter paper (8 μm, No. 540, Whatman Ltd, England, UK). The filter paper was washed thoroughly with hot water (80 °C) until free from sugar before being dried for 60 min at 130 °C, cooled and re-weighed. The difference between prefiltered and postfiltered weight was divided by the mass of honey and expressed as percent water-insoluble solid.

Free acidity was determined by dissolving 10 g of honey in 75 ml of distilled carbon dioxide-free water. The sample solution was titrated against 0.1 N sodium hydroxide solution using 4–5 drops of phenolphthalein indicator (*Bogdanov, Martin & Lullmann, 2002*). The result was expressed as milli-equivalents acid/kg honey.

Diastase content was determined by following the Phadebas® Honey Diastase Test. In brief, exactly 1 g of honey was dissolved in 100 ml acetate buffer solution. A five mL sample of this solution was incubated at 40 °C and, after 5 min, one Phadebas tablet was added

and vortex mixed. The solution was then incubated at 40 °C for 30 min before adding one mL of 0.5 M sodium hydroxide. The mixture was vortexed and centrifuged at 1,500 G for 5 min before its absorbance was measured at 620 nm (Cary 60 UV-Vis; Agilent, Santa Clara, CA, United States) (*Phadebas, 2024*; *Tosi et al., 2008*). A blank without honey was treated in the same manner. The result was expressed as a Diastase Number (DN) from the chart listed in the Phadebas® Honey Diastase Test user manual.

The 5-Hydroxymethylfurfural (HMF) content of the honey was determined following the White method (*White, 1979*). In brief, exactly 5 g of honey was dissolved in 25 mL of deionized water. The solution was transferred into 50 mL volumetric flasks and 0.5 mL of Carrez Solution I was added. After mixing, 0.5 mL of Carrez Solution II was added, followed by a thorough mixing. The volumetric flask was made up to 50 mL with deionized water and filtered (Grade 4: 20–25 $\mu$m; Whatman Ltd, Little Chalfont, UK). Two samples of five mL of filtrate each were collected in two different test tubes. five mL of deionized water was added to one of the test tubes (sample), whereas five mL of 0.20% sodium bisulfite (NaHSO$_3$) solution was added to the second test tube (reference). Both test tubes were vortex mixed and the absorbance of the samples measured at 284 nm and 336 nm (Cary 60 UV-Vis; Agilent) with the sample compared against the reference as blank (*PerkinElmer, 2023*).

The HMF content of honey was calculated using following equation:

$$HMF\left(mg/100\ g\ Honey\right) = \frac{(A_{284} - A_{336}) \times 74 \cdot 87}{w}$$

Where, w = weight of sample (g)

The HMF content was expressed as mg/kg honey.

## RESULTS AND DISCUSSION

### HPTLC analysis of organic honey and nectar extracts

Over five years, more than 500 honey samples from the southwest of Western Australia were collected, many of them Jarrah as per beekeeper identification. To determine the common characteristics of Jarrah honey, this article refers to 31 beekeeper-identified Jarrah honey samples. These were extracted, and their organic extracts analysed by HPTLC (see Method 'Jarrah nectar and jarrah honey organic extracts'). HPTLC results were obtained under four different light conditions (at 245 nm and 366 nm developed; under white light and at 366 nm derivatised). The HPTLC signatures of each investigated Jarrah honey (Fig. S1) revealed a common, unique, banding pattern through the Rf value of individual bands, as well as the band colour and intensity under the different light conditions. It was presumed that the common HPTLC signature was reflective of that of a 'typical' Jarrah honey.

To confirm this proposed Jarrah honey signature, 5 $\mu$L of Jarrah flower nectar extract was also analysed using the same HPTLC method. A comparable signature was confirmed (Fig. 2) with key bands previously identified in the individual Jarrah honey extract samples also present in Jarrah nectar extract.

With the proposed HPTLC signature of Jarrah honey (Table 2, Fig. S1) further supported by the analysis of Jarrah flower nectar (Table 3, Fig. 2), the individual signatures of each
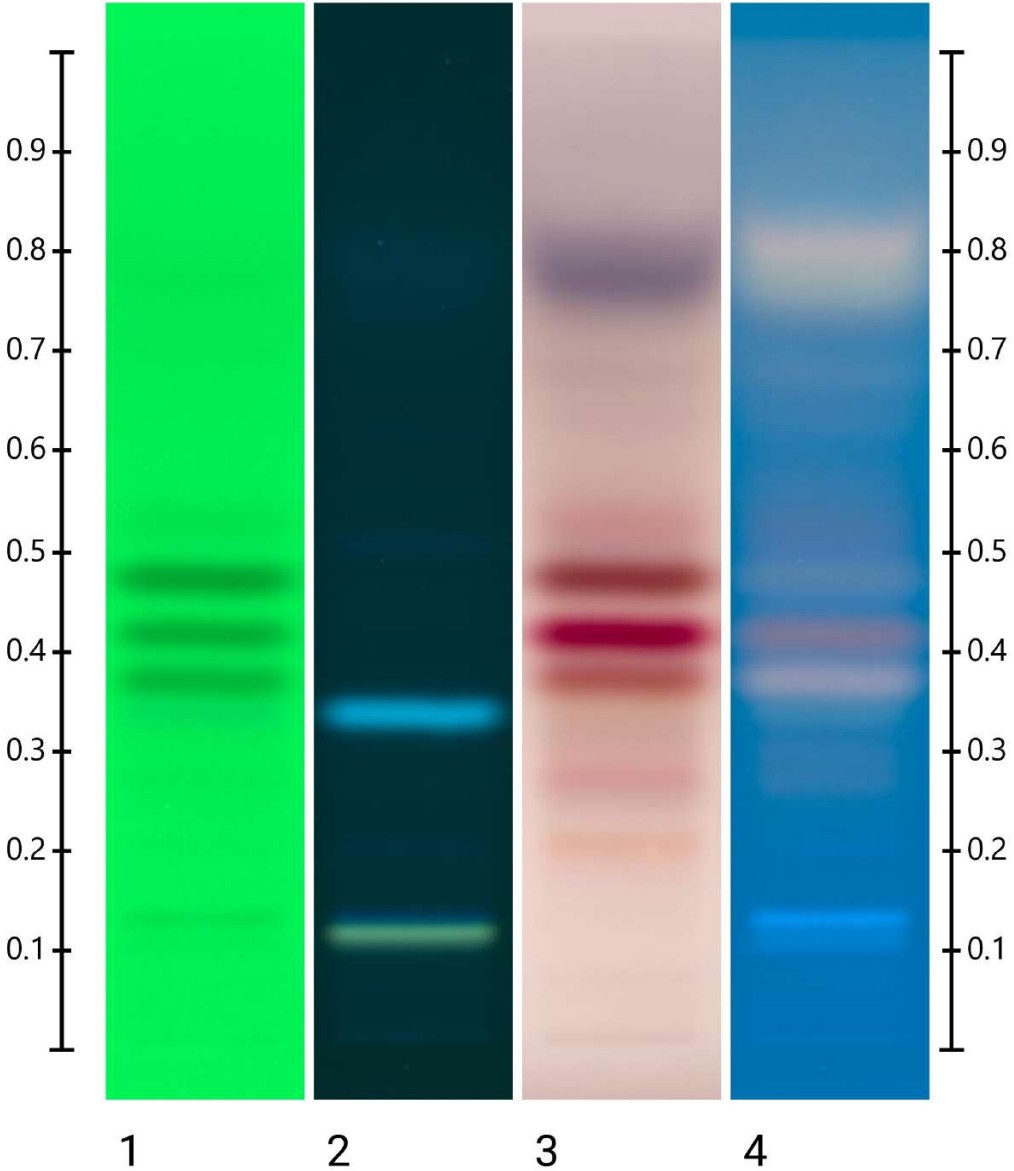

**Figure 2  HPTLC image of 5 μL of Jarrah nectar extract.** Images were taken before derivatisation at 254 nm (Track 1) and 366 nm (Track 2); and after derivatisation with vanillin reagent at white (Track 3) and 366 nm (Track 4).

of the 31 investigated beekeeper-identified Jarrah honey samples were once more closely inspected. Some were found to display additional bands, while others were paler due to the presence of another nectar source with lower levels of extractable compounds (*e.g.*, JAR –66, JAR –266, JAR –287, JAR –367, JAR –402). Both findings suggest co-flowering. After careful visual analysis, samples JAR –75, JAR –172, JAR –258, JAR –265, JAR –266, JAR –295, JAR –302, JAR –313, JAR –328, JAR –338, JAR –342, JAR –344 and JAR –373 were selected as they had strong HPTLC Jarrah signatures and were identified as being free of noticeable additional bands resulting from co-flowering. These select samples were considered the

**Table 2** **The dominant common bands found from the HPTLC analysis of the organic extract of Jarrah honey at 254 nm and 366 nm and derivatised at white light and 366 nm proposed as the Jarrah honey signature.** Colours in bold indicate dominant bands of colour.

| Visualisation | 254 nm (Green) | 366 nm (Black) | Derivatised white light (White) | Derivatised 366 nm (Blue) |
|---|---|---|---|---|
| **Rf** | | | | |
| 0.11 | | Yellow | | Light Blue[*] |
| 0.22 | | | **Dark Brown** | Beige |
| 0.32 | Blue | **Bright Blue** | Pale Blue | Light blue |
| 0.33 | **Pale Black** | | | |
| 0.38 | | | Light Pink | **Light Pink** |
| 0.41 | **Black** | | **Red** | **Orange Pink** |
| 0.47 | **Pale Black** | | Pale Blue | **Pale Blue** |
| 0.57 | | | | Pale Blue |

**Notes.**
*Often seen as a double band.

**Table 3** **Jarrah flower nectar bands in the organic extract present at 254 nm and 366 nm and derivatised at white light and 366 nm.** Colours in bold indicate dominant bands of colour.

| Visualisation | 254 nm (Green) | 366 nm (Black) | Derivatised white light (White) | Derivatised 366 nm (Blue) |
|---|---|---|---|---|
| **Rf** | | | | |
| 0.11 | | **Yellow** | | Bright Blue |
| 0.22 | | | **Orange** | |
| 0.32 | Blue | **Bright Blue** | | |
| 0.33 | | | | |
| 0.38 | **Black** | | **Brown** | **Light Pink/Yellow** |
| 0.41 | **Black** | | **Red** | **Orange Pink/Yellow** |
| 0.47 | **Black** | | **Brown** | Beige/Pink |
| 0.57 | | | | |

purest representations of Jarrah honey and were blended by mixing equal amounts to yield a representative Jarrah sample for subsequent HPTLC and physicochemical analysis. The concept of a blended sample is that, despite some natural variations within each sample, dominant features can be discerned and therefore 'typical' characteristics, for example, common physicochemical parameters or a representative HPTLC signature, can be derived (*Islam et al., 2021a*).

When placing the HPTLC fingerprint of the organic extract of the blended sample (Fig. 3A) alongside the Jarrah nectar extract signature (Fig. 3B), though varying in intensity, bands are common to both the Jarrah honey and Jarrah flower nectar. Dominant bands previously seen in all individual Jarrah honey samples (Supplementary data and Table 2) are also present in the blended sample (Fig. 3A), confirming these bands form a typical HPTLC signature.

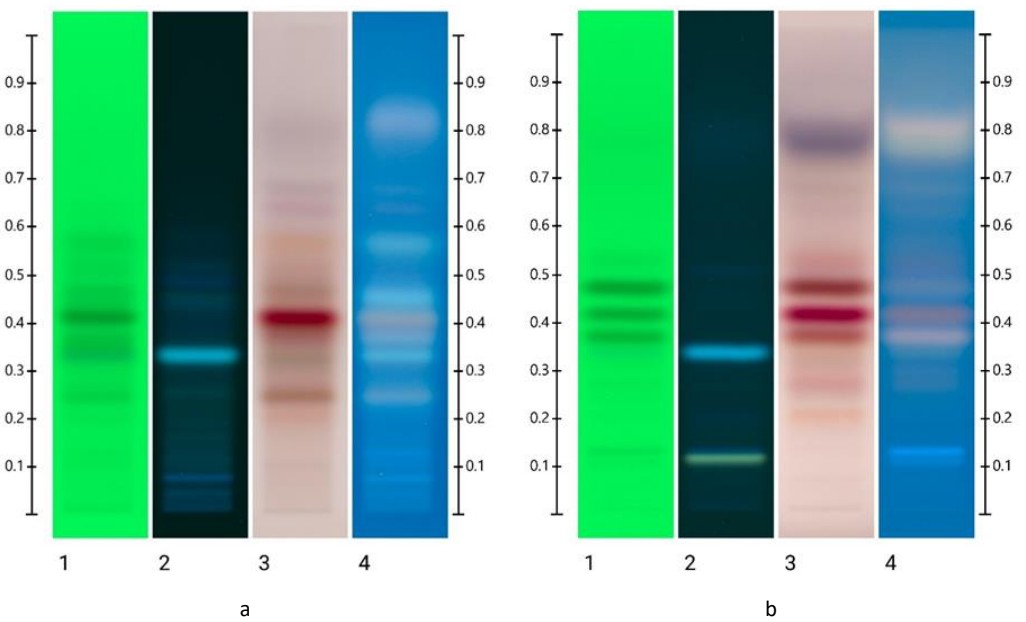

**Figure 3** **HPTLC images of (A) blended Jarrah extract; (B) the nectar extract.** Images taken at (Track 1) 254 nm and (Track 2) 366 nm; and after derivatisation at (Track 3) white light and (Track 4) 366 nm.

## Physicochemical properties of Jarrah honeys

The amount of water in honey determines its ability to resist spoilage by yeast fermentation and to remain stable during storage (*Umesh Hebbar, Rastogi & Subramanian, 2008*). A high moisture content can affect the physical properties of honey, including its viscosity and tendency to crystallise. The moisture content of the blended Jarrah honey sample was 16.8%, well within Codex Alimentarius requirements (Table 4), indicating a high-quality honey that will store well.

As per the analysis of the blended sample, the predominant sugars in Jarrah honey are fructose (42.5 g/100 g) and glucose (20.8 g/100 g) (Fig. 4). Maltose is also present in measurable amounts (1.9 g/100 g), while the level of sucrose is below 0.5 g/100 g. These levels all fall within Codex Alimentarius guidelines for honey. Previous studies (*Dawes & Dall, 2014*; *Manning, 2011*) on Jarrah honey have also reported significantly higher fructose than glucose levels (almost double), which contributes to the typical slow crystallisation tendency (*Pita-Calvo, Guerra-Rodriguez & Vazquez, 2017*) that Jarrah honey is known for.

Any insoluble matter present in honey, including pollen, honeycomb debris, bee, and filth particles, is a crucial criterion for determining honey cleanliness. Unfiltered or raw honey usually has a higher water-insoluble content with an acceptable amount as per Codex Alimentarius being below 0.1% (*Alimentarius, 2017*). The blended Jarrah honey sample had a water-insoluble content of 0.04% indicating that the individual Jarrah honey samples used to prepare the blend had a high level of cleanliness.

High levels of free acidity in honey can indicate that the honey has undergone fermentation by yeast. During this process, glucose and fructose are converted into

**Table 4** Physicochemical characteristics of blended Jarrah honey with reference to Codex Alimentarius guidelines for honey.

| Revised codex standard for honey (*Alimentarius, 2017*; *Bogdanov, Martin & Lullmann, 2002*) | | Blended Jarrah honey |
|---|---|---|
| **Test** | **Accepted range** | |
| Moisture | Not more than 20% | 16.8% (Brix 81.9%) |
| | Sugar content | |
| Fructose and glucose (sum of both) | Not less than 60 g/100 g | Fructose 42.49 and Glucose 20.81 Sum 63.3 g/100 g |
| Sucrose | Not more than 5 g/100 g (*Eucalyptus camaldulensis* is given an exception of 10 g/100 g) | Below 0.5 g/100 g |
| Water insoluble solids content | Non-pressed honey is not more than 0.1 g/100 g | 0.04 g/100 g |
| Free acidity | Not more than 50 milliequivalents acid per 1,000 g | 19 milliequivalents acid per 1,000 g honey |
| Diastase activity after processing | Not less than 8 Schade units, and in the case of honeys with a low natural enzyme content, not less than 3 Shade units | 13.2 Diastase Number |
| Hydroxymethylfurfural content after processing | Not more than 40 mg/kg | 20.36 mg/kg |
| Electrical conductivity | Eucalyptus honey is an exception to the standard of not more than 0.8 mS/cm | 1.31 mS/cm |

ethanol and carbon dioxide. In the presence of oxygen, ethanol is further transformed into acetic acid, which raises the free acidity of the honey (*Ajlouni & Sujirapinyokul, 2010*). For the blended Jarrah honey a free acidity of 19 milli-equivalents acid per kilogram of honey was determined, which is well below Codex Alimentarius requirements of less than 50 milli-equivalents acid per kilogram of honey.

Whilst not included in the Codex standard for honey, the pH of the blended Jarrah sample was also measured and found to be 4.95. Honey is naturally acidic with its pH typically ranging between 3.42 to 6.10 (*Yang et al., 2019*). A previous study suggested that the pH of Jarrah was between 4.40 to 5.70 (*Dawes & Dall, 2014*), which is in line with the findings reported here.

Diastase, also known as amylase, is a naturally occurring enzyme in honey. Its main function is to break down starch into short-chain sugars. The level of activity of this enzyme in honey can be an indicator of the quality of storage conditions and possible heating processes the honey has undergone. The Diastase Number for the blended Jarrah honey sample was 13.2 DN which was well above Codex standard levels (above 8 Schade units) (*Alimentarius, 2017*).

Hydroxymethylfurfural (HMF) is a natural by-product of sugar decomposition, primarily fructose (*PerkinElmer, 2023*; *Zappalà et al., 2005*). It is formed slowly due to the acidic pH of honey, but heat treatment can accelerate its formation. Despite its

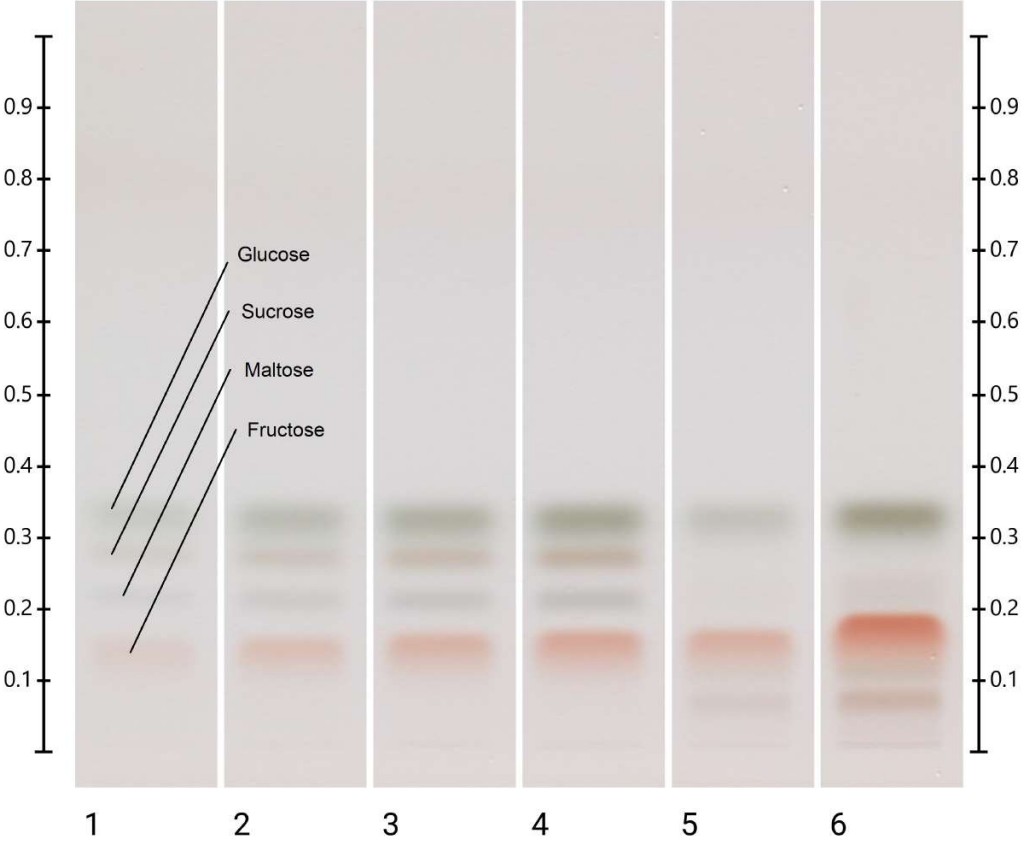

**Figure 4** **HPTLC images taken at White light after derivatisation; Track 1–4—Standards (fructose, maltose, sucrose, and glucose in 1–4 µL), Track 5 —Jarrah (2 µL) and Track 6 —Jarrah (6 µL) of aqueous methanolic honey solution.**

relatively high fructose content, HMF levels in the blended Jarrah honey were found to be 20.36 mg/kg, which is well within Codex Alimentarius guidelines (*Alimentarius, 2017*).

Honey contains certain minerals that contribute to the ash content of honey, and this together with its acid content leads to an increase in electrical conductivity. Typically electrical conductivity ranges between 0.1 and 3 mS/cm (*Bogdanov, Martin & Lullmann, 2002*); however, the Codex Alimentarius for honey generally requires an electrical conductivity of less than 0.8 mS/cm with Eucalyptus honey being an accepted exception (*Alimentarius, 2017*). As Jarrah is a Eucalypt, the electrical conductivity of the blended Jarrah honey sample was found to be 1.31 mS/cm, which is in line with previous studies reporting 0.716–1.157 mS/cm (*Dawes & Dall, 2014*) and 1.39–1.60 mS/cm (*Islam et al., 2022*).

## Identification of marker compounds in Jarrah honey

To further strengthen authentication and quality control efforts for Jarrah honey, chemical identification of the compounds constituting the identified HPTLC signature, was attempted. The approach taken was based on a method developed by *Lawag et al.*

**Table 5 Six dominant bands detected in organic extracts of both Jarrah nectar and Jarrah honey at 254 nm and 366 nm and derivatised at white light and 366 nm.** Colours in bold indicate dominant bands of colour.

| Bands (Rf) | 254 nm (Green) | 366 nm | Derivatised white light (White) | Derivatised 366 nm (Blue) | Possible phenolic identification |
|---|---|---|---|---|---|
| 0.11 | | Yellow | | Light Blue | epigallocatechin gallate (EGCG) |
| 0.22 | | | **Dark Brown** | Beige | unknown |
| 0.32 | Blue | **Bright Blue** | Pale Blue | Light blue | lumichrome |
| 0.33 | **Pale Black** | | | | |
| 0.38 | | | Light Pink | **Light Pink** | taxifolin |
| 0.41 | **Black** | | **Red** | **Orange Pink** | unknown |
| 0.47 | **Pale Black** | | Pale Blue | **Pale Blue** | o-anisic acid (O-AA) also known as 2-methoxybenzoic acid |
| 0.57 | | | | Pale Blue | |

*(2023)* where key HPTLC parameters of individual bands before and after derivatisation (*i.e.,* Rf, RGB value, UV and fluorescence spectral data) were matched against a comprehensive database of (mainly phenolic) compounds that were previously reported as honey constituents. The following compounds were identified as likely matches for the dominant bands in the Jarrah honey and nectar HPTLC signature: epigallocatechin gallate, lumichrome, taxifolin, o-anisic acid (2-methoxybenzoic acid), kojic acid, hesperidin, m-coumaric acid and 2,3,4-trihydroxybenzoic acid. These were linked to the bands detected in the organic extract of Jarrah honey (Table 5).

To confirm the tentative identification of key bands standards of epigallocatechin gallate, lumichrome, taxifolin, o-anisic acid (2-methoxybenzoic acid), kojic acid, hesperetin, m-courmaric acid and 2,3,4-trihydroxybenzoic acid, these were run alongside Jarrah honey and nectar extracts using the previously described HPTLC method (section 2.6.1). The compound identification was confirmed by adopting a spectral matching approach (*Lawag et al., 2022*) where UV-Vis spectra (200–850 nm) of the bands of interest in Jarrah honey and nectar extracts were compared with those of the respective standards. An illustrative example using lumichrome (Fig. 5), presents the UV-Vis spectra of this standard alongside the UV-Vis spectra of the corresponding band in Jarrah honey and nectar extracts (Rf 0.32). Spectral comparisons for epigallocatechin gallate, taxifolin, and o-anisic acid are included in Fig. S1 (Figs. S2–S6).

Two bands in the HPTLC signature of Jarrah honey and nectar (Rf 0.22 and 0.41) remain unidentified using this compound identification approach. Future studies, possibly relying on other instrumentation, are necessary to confirm the identity of these two constituents.

## CONCLUSION

With the increase in demand for Jarrah honey as a premium product, beekeepers require chemical traceability to demonstrate the authenticity of this unique honey from the

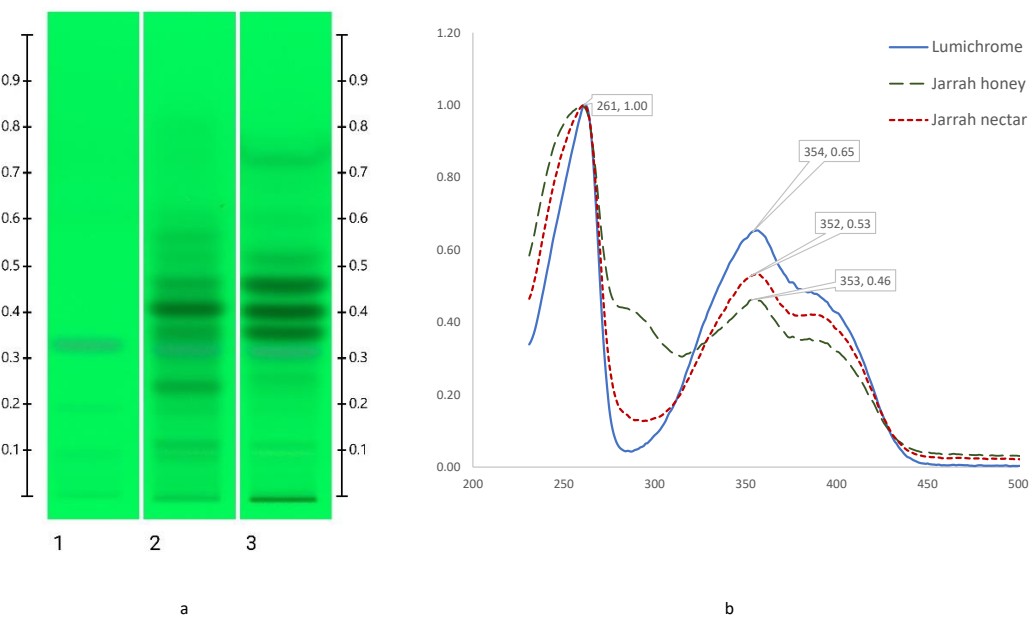

Figure 5  HPTLC images at 254 nm (A) and spectral analysis of band at Rf 0.32 (B); Track 1 – Lumichrome, Track 2 –Jarrah Honey and Track 3 –Jarrah Nectar.

southwest of Western Australia. This is a common need for the protection of exceptional monofloral honey types worldwide. Governing bodies are searching for methodologies to establish new standards by which they can prove honey authenticity and prevent fraudulent activities of adulteration or mislabelling of the product. This article outlined a new approach to monofloral honey authentication using a unique HPTLC signature derived from the honey's organic extract.

The advantage of this approach is that an HPTLC-derived signature of the organic extract, unique to each honey type, can be established and visually interpreted based on typical banding patterns (*i.e.,* RF values) and respective band colours (*i.e.,* RGB values converted into colour hues). This enables the use of this signature for honey-type authentication without complete chemical identification of each of the bands. Compound identification, as has already been undertaken as part of this study, will further assist any authentication efforts as well as provide insight into the potential effects of honey ripening on its chemical composition compared to the nectar resource.

It has been suggested that melissopalynology is inaccurate for the identification of some monofloral honey types from Australia (*Islam et al., 2022*). Commonly beekeeper businesses are migratory moving great distances to access the flowering cycle of the region which, in the southwest of Western Australia, is renowned for its rich biodiversity. Pollen contamination of the honey can originate from a wide source of plants making melissopalynology an inadequate approach for monofloral honey confirmation in the Australian landscape. A melissopalynology study of Australian honey types (*Sniderman et al., 2018*) concluded that the existing International Honey Commission criteria for authenticating Eucalyptus honey should not be relied upon for Australian honey, since

those criteria are not based on samples of Australian honey and are often taken from single-species plantings. The HPTLC-based approach proposed in this study which analyses the honey's organic extract, offers an alternative authentication approach and may provide a more accurate tool for the determination of honey type.

Since honey is produced from honeybee-collected and ripened flower nectar or plant exudates, the presence of nectar-derived compounds in the honey is direct evidence of meeting the Codex Alimentarius definition for honey and provides a direct chemical link to the plant species' nectar source. The use of phenolics to identify the nectar source in honey is not a novel concept (*Ferreres et al., 1996*; *Nešović et al., 2020*). *Baker & Baker (1983)* discovered that phenolic-rich nectars are present in 333 out of 850 plant species they tested, indicating their widespread occurrence in nature. The reason for their presence has been extensively debated with suggestions including their ability to fluoresce to act as a feeding guide (*Thorpe et al., 1975*); their taste acting as a deterrent to nectar-robbing and an attractor to pollination (*Hagler & Buchmann, 1993*); or their ability to act as an antimicrobial defence (*Adler, 2003*). These phenolics appear to increase in their dominance when moving from tropical to harsher climatic regions (*Baker & Baker, 1983*). The documented importance of phenolics in flower nectar, and by extension therefore also in honey, supports the use of an organic honey extract to confirm its respective nectar source. The HPTLC-derived signature presented in this study offers a new approach to authenticating honey for consumer protection. Identification of these compounds further clarifies the link between nectar and honey and may also provide additional insight into their function.

Confirming the honey type enables its accurate physiochemical characterisation. This can be used as supporting evidence of the honey types specific nectar source (*i.e.,* sugar profile and electrical conductivity) and handling of the honey during extraction and storage processes (*i.e.,* moisture content, insoluble matter, free acidity, diastase activity and HMF) and further assist quality control efforts by the honeybee industry.

## ACKNOWLEDGEMENTS

Honey samples were supplied by the Beekeeping Industry Council of Western Australia (BICWA).

### Funding

This research was supported by funding from the Cooperative Research Centre for Honey Bee Products and its spinout company, Y-Trace Pty Ltd. The funders had no role in study design, data collection and analysis, decision to publish, or preparation of the manuscript.

### Competing Interests

The authors declare there are no competing interests.

## Author Contributions

- Md Khairul Islam conceived and designed the experiments, performed the experiments, analyzed the data, prepared figures and/or tables, authored or reviewed drafts of the article, and approved the final draft.
- Elizabeth Barbour conceived and designed the experiments, performed the experiments, prepared figures and/or tables, authored or reviewed drafts of the article, review and editing, and approved the final draft.
- Cornelia Locher conceived and designed the experiments, authored or reviewed drafts of the article, review and editing, and approved the final draft.

## Data Availability

The data are available in the figure, tables, and Supplementary File.

## Supplemental Information

Supplemental information for this article can be found online at http://dx.doi.org/10.7717/peerj-achem.33#supplemental-information.

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
