# Peer review of "Authentication of Jarrah (*Eucalyptus marginata*) honey through its nectar signature and assessment of its typical physicochemical characteristics"

_PeerJ Analytical Chemistry, doi:10.7717/peerj-achem.33_

## Round 0.1 · original submission · Major Revisions

Reviewers have raised major concerns on your submission.

Please carry out the requested revisions. Where you are not in total agreement, kindly provide a rebuttal letter explaining your reasons.

I look forward to receiving your revised manuscript.

With kind regards,
Timothy Omara

·

Basic reporting

The manuscript is generally well written with few grammatical or typo mistakes. I do have some enquiries over references though.
Line 48: reference (Specht) without year
Line 58: reference (2002) without author/editor
Line 107: (2012) without author/editor
Line 108: (A.V. Slee) without year
Line 129-130: To this reviewer, it appears if BOTH honey and nectar solutions were diluted with 2 mL deionised water. Is this correct?
Line 151: typo, zinc acetate is soluble to 30 g per 100 mL water
Line 208: reference (2024) without author/editor
Line 266: Some were found to display additional bands while others were paler …
Line 350: why [40]?
Line 353: Is [39] referring to a reference?
Line 357: Should (Table 4 and 5) be (Table 5) only?
Line 390: taxifolin, not taxifoline
Line 395-396: The quoted fructose plus glucose, and fructose do not match the quantities quoted in Table 4.

Overall, the manuscript is well communicated and very readable.

Experimental design

The experiments are clear and appropriate, except for the two enquires.

Validity of the findings

The findings are justified and valid. The authors did not speculate wildly on the “unknowns”. Instead, they stayed true to the manuscript title stuck to what could be proven.

Additional comments

Enquiry to authors, could a difference in nectar verse honeys bands be due the nectar contains glycosides and the honey aglycones?

This manuscript reports on the authentication of a high value honey by HPTLC. This novel technique tracks nectar signature bands to honey signature bands. I anticipate that it could equally be adapted to other high value honeys.

As such, the findings and protocols have implications beyond jarrah honeys and will be of general interest to the honey industry.

I recommend publication after a few minor review enquires are cleared up.

Reviewer 2 ·

Basic reporting

English is ok.
Literature is sufficiently sampled.
The structure of the article is ok.

Experimental design

The experimental design is not complete.
- A first question is concerning the evaluation of authentic Jarrah honey and nectar from other geographical origins (whenever possible). How the authors evaluated this point?
- Additionally, did the authors evaluated mixtures of honey from WA and Jarrah to evaluate the LOD of the approach for detecting mixtures/dilutions practices? Eventually other non-WA honeys.
- In the present work the authors evaluated different HPTLC bands attributed to co-flowering, therefore blended samples. How is the allowed limit of blends for a pure Jarrah honey?
- Which are the values of reproducibility and accuracy of the rf values of bands for the proposed HPTLC approach?
- How many replicates have been considered in this study?

Validity of the findings

The novelty is not present.
The approach includes an HPTLC analysis combined with physicochemical characterizations. The authors also stated that this authentication approach allows, for the first time, the characterization of physicochemical characteristics of this honey. The present manuscript is an extension of a previous work from the same group, in which the same approach was applied to a restricted samples set of Jarrah honey (https://doi.org/10.1016/j.crfs.2022.02.014), while 31 were considered here.

Additional comments

The proposed approach resulted fruitful in determining compositional differences, typically in botanical assessments, or in the profiling of polyphenols compositions etc.. In fact the authors already published compositional differences between Manuka and Jarrah honeys in their recent work (https://doi.org/10.7717/peerj.12186). The main result of these investigations is concerning a qualitative evaluation of HPTLC bands, whereas the quantitative evaluation (e.g. saccharides content) could be achieved by other methods. The authentication process needs more accurate determinations.

·

Basic reporting

Comments
The developed method for the authentication of unifloral honey (Jarrah), is novel. The concern for botanical authentication of honey is global and present methods need many specifications that is significantly covered by the researchers of this article. Moreover, this study will help other researchers to authenticate the different unifloral verities across the globe. After critically reviewing this paper I accept this manuscript for the publication, but before that there are certain points that need improvement.

Experimental design

The following points need much consideration by the author to improve the quality of this paper.
1. In the abstract section line number 28-29, the meaning of the sentence is unclear, please make it clearer.
2. In line number 63-65, add European stander for monofloral honey (pollen percentage).
3. Line number 65-63 add average glycaemic index of jarrah honey.
4. Line number 82-84: The conclusion part of introduction section needs much elaboration, that who this method will help the governing bodies to establish new standers for unifloral honey and tackle the issue of unethical labelling of unifloral honey.
5. In section 2.2.1, clearly mention proper location with GPS coordinates and the date when the nectar sample were collected.
6. In section 2.2.3 the preparation of jarrah honey blend is unclear, elaborate this section so that the read didn’t get confused.
7. In the results section the author refers 31 honey samples (section 3.1) and the authentication procedure like melissopalynology is missing. Basically, the team is trying to develop an authentication method but there is a lack of validation part with existing methods. So, it is a advice to look into this matter. Because at this point before developing a method you should collect the data based on existing methods. How can you relay on the beekeepers without analysing the facts. The analysed samples might be not under the EC standers for unifloral honey (>45% pollen of one source)..
8. Author has to add the microscopic images of Jarrah pollen.
9. In Results and discussion section 3.2. The author has collected the samples from 2015-2020 (table 1). And it is evident that the storage time has greater effect of the DN and HMF content as well as other compounds in the honey. In the paper on one value of each parameter is mentioned. Please mention every parameter of each sample in the table only and elaborate the discussion (considering the moisture, Diastase and HMF) to make it understandable to the audiences.
10. Line number 348 need correction, rewrite the sentence.
11. Line 350 and 365 citations need correction.

Validity of the findings

12. The conclusion part is lacking in defining the scope of the proposed method, so improve the conclusion and hight the scope and applications. Overall, the structure of conclusion need refinement (general statement, aim, finding, scope, conclusion highlights the benefits of research to the society).

After all these corrections the paper can be accepted for publication.

---

## Round 0.2 · accepted · Accept

Dear Authors,

Thank you revising your manuscript following the reviewer suggestions

·

Basic reporting

This first revision is significantly improved from the original manuscript.
Only one minor typo in line 414 of the clean copy, "pollinatingn"

Experimental design

Experiments are relatively novel for honey authentication, and methods are valid.

Validity of the findings

Findings are all valid.

Additional comments

All recommendations from the original review have been address and the general readability has been improved.

Reviewer 2 ·

Basic reporting

The revised manuscript has been proposed according to the reviewer questions arose, even though some were rejected.

Experimental design

nothing to comment

Validity of the findings

nothing to comment

Additional comments

nothing to comment

·

Basic reporting

after reading the revised manuscript I am satisfied with all the changes done by the author, and accept the paper in its current form

Experimental design

all the changes are accepted

Validity of the findings

all the changes are accepted

Additional comments

all the changes are accepted and I am satisfied with all the changes done by the author, and accept the paper in its current form